# Microembolizations in the Arterial Cerebral Circulation in Patients with Atrial Fibrillation Ablation Using the Cryoballoon Technique—Protocol and Methodology of a Prospective Observational Study

**DOI:** 10.3390/diagnostics13091660

**Published:** 2023-05-08

**Authors:** Damir Erkapic, Marko Aleksic, Konstantinos Roussopoulos, Kay Felix Weipert, Korkut Sözener, Karel Kostev, Jens Allendörfer, Josef Rosenbauer, Dursun Guenduez, Christian Tanislav

**Affiliations:** 1Department of Cardiology, Rhythmology and Angiology, Diakonie Klinikum Jung Stilling, 57074 Siegen, Germanyjosef.rosenbauer@diakonie-sw.de (J.R.);; 2Department of Cardiology and Angiology, University Clinic of Giessen, 35392 Giessen, Germany; 3Department of Rhythmology, Klinikum Hanau, 63450 Hanau, Germany; 4Epidemiology, IQVIA, Unterschweinstiege 2-14, 60549 Frankfurt am Main, Germany; 5Neurological Clinic Bad Salzhausen, 63667 Nidda, Germany; 6Department of Geriatrics and Neurology, Diakonie Klinikum Jung Stilling, 57074 Siegen, Germany

**Keywords:** atrial fibrillation, cryoballoon, ablation, microembolization, neuropsychological outcome

## Abstract

There is considerable uncertainty regarding the impact of microembolic signals (MESs) on neuropsychological abilities in patients receiving pulmonary vein isolation and beyond using the cryoballoon technique. We conducted the largest prospective observational study on this topic, providing insights into the gradual unmasking of procedure-related MESs and their impacts on neuropsychological outcomes. MESs were continuously detected periprocedurally using transcranial Doppler ultrasonography. Neuropsychological status was evaluated comprehensively using the CERAD Plus test battery, which consists of 11 neuropsychological subtests. Patients with atrial fibrillation were included in the study with an equal distribution (50:50) of paroxysmal or persistent presentations. Of 167 consecutive eligible patients, 100 were included within the study enrollment period from February 2021 to August 2022. The study, including the documentation of all follow-up visits, ended in November 2022. This paper focuses on describing the study protocol and methodology and presenting the baseline data.

## 1. Introduction

Atrial fibrillation is the most common form of cardiac arrhythmia in adults [1]. The incidence of this condition increases with age, and it is associated with increased morbidity and mortality [1,2,3,4,5]. Catheter-based ablation of atrial fibrillation is an established method for the treatment of atrial fibrillation [6]. In addition, recent studies have shown that ablation therapy using the cryoballoon technique is a superior first-line therapy to antiarrhythmic medication in terms of its effectiveness in preventing recurrences [7,8,9,10]. As an add-on to pulmonary vein isolation alone, some authors have reported cryoballoon left atrial roof ablation as being beneficial for achieving a better arrhythmia-free outcome in patients with persistent atrial fibrillation [11,12,13]. With regard to the safety aspects of atrial fibrillation ablation, it is known that silent microembolic events occur in up to 50% of all patients with pulmonary vein isolation, depending on the ablation technology and strategy used [14,15,16,17]. Although the occurrence and number of silent microembolizations during different cardiac interventions suggest clinical implications, their significance regarding the short- and medium-term neuropsychological abilities of patients after atrial fibrillation ablation is ultimately unclear [18,19,20]. Most studies that have examined this question regarding atrial fibrillation ablation only included a small number of cases and were based on a heterogenous patient collective with regard to different stages of atrial fibrillation and the simultaneous use of different ablation energy sources [19,20,21,22,23]. Furthermore, no data are available concerning the microembolic burden, characteristics, and enabling factors in patients receiving extrapulmonary ablation using the cryoballoon technique.

In view of this, we conducted the largest prospective observational study on this topic to date, providing insights into the gradual unmasking of procedure-related microembolic signals (MESs) and their impacts on neuropsychological outcomes.

## 2. Materials and Methods

Patients with atrial fibrillation, who were referred to the Atrial Fibrillation Center Diakonie Klinikum Jung Stilling in Siegen, Germany for ablation therapy, were included in this study. The study protocol was approved by the Westfalen–Lippe ethics committee of the University of Münster, Germany (AZ 2019-779-f-S). All patients provided signed informed consent forms prior to enrollment. Patient data were collected and stored in an anonymized fashion in compliance with the requirements set forth in the applicable data protection laws.

### 2.1. Patient Population

Patients with paroxysmal or persistent atrial fibrillation—defined on the basis of the current guidelines [6]—were eligible to participate in this study if they met the inclusion criteria displayed in Table 1. Patients were excluded if any of the following applied to them: (1) age ≤ 18 years; (2) permanent atrial fibrillation; (3) interventional coronary intervention and/or coronary artery bypass surgery with or without myocardial infarction within the 3 months prior to ablation therapy; (4) intracardiac thrombus; (5) contraindication for oral anticoagulation; (6) chronic kidney disease requiring dialysis; (7) pregnancy; (8) mitral valve regurgitation > grade 2; (9) left atrial diameter ≥ 60 mm; (10) known cognitive impairment (e.g., dementia); (11) simultaneous participation in another study; (12) lack of a transcranial bone acoustic window; (13) presence of other left atrial arrhythmias in addition to atrial fibrillation; and (14) if patient did not expect to be available for the three-month follow-up appointment.

### 2.2. Study Design and Objectives

The study was designed as a prospective observational study. The study steps are summarized in the study flowchart (Figure 1). Upon study inclusion, patient characteristics, comorbidities, and drugs were recorded (Table 2). Depending on the stage of atrial fibrillation, patients were assigned to one of two groups (Figure 2).

In line with the internal standard procedure, all patients scheduled for an ablation were treated as described above. This approach was established on the basis of the current evidence surrounding ablation and its effects. A number of reports in the literature suggest that patients with persistent atrial fibrillation might benefit more from the application of additional atrial roof ablation [11,12,13]. The procedural approach used with patients in our study was based on an internal protocol in view of such findings, rather than a component of the study protocol. In this context, the attempt to include two groups of equal size with paroxysmal and persistent atrial fibrillation was intended to homogenize the study population. This was one of the points made concerning previous studies, in which colleagues investigated randomly heterogenous populations with consecutive patients [19,20,21,22,23].

The main purpose of the present study was to investigate the burden of cerebral MESs during the procedure. The study was conducted using a cross-sectional design. Using this as starting point, we investigated the influence of the ablation procedure on the cognitive ability of each individual patient for the entire cohort (longitudinal cohort study design). The main objective was to determine whether the procedure generally results in a cognitive decline proximate to the procedure and at a medium-term follow-up point. Here, the hypothesis was that the procedure would affect the cognitive performance. As a subgroup analysis, it is beneficial to analyze the effects separately in the subgroups of persistent and paroxysmal AF patients; in our study, the number of included patients per subgroup actually surpasses the numbers in previous studies addressing this topic. In a further step, we aimed to demonstrate a potential correlation between cognitive decline and the burden of MESs.

As an integral part of the study protocol, MESs were continuously recorded during all ablation procedures in each patient using transcranial Doppler ultrasonography. Furthermore, within 24 h prior to ablation therapy, all patients underwent an extensive neuropsychological examination using the CERAD Plus battery, which consisted of 11 neuropsychological tests. The neuropsychological examination was repeated within 24 h after the ablation procedure and again 3 months after.

Periprocedurally recorded data are displayed in Table 3. Follow-up data were recorded up to three months after the ablation procedure and are shown in Table 4.

The primary objectives can be summarized as follows:

(1) determination of the frequency of MESs in the cerebral circulation during the procedure (cross-sectional design); (2) assessment of the potential cognitive decline caused by the procedure (longitudinal cohort study design); (3) analysis of the potential causative correlation between the frequency of MESs and cognitive decline. As the data set created allowed for further analyses, we also defined secondary objectives, such as the identification of factors potentially associated with cognitive decline after the procedure, the burden of MESs, and the occurrence of adverse events during the procedure.

The potential technical and individual influencing factors examined in this study were:
(1)Technical variables that relate to the electrophysiological procedure (venous access, trans-septal puncture, pulmonary vein/left atrium angiography, cryoballoon introduction into the left atrium, balloon- and contrast-agent-guided pulmonary vein occlusion, balloon freeze cycle and thawing phase for each PV, LARA freeze cycles, and thawing phase and sheath retreat to the RA);(2)Individual factors:
(a)Investigator dependency (effect of the examiner’s experience in cryoballoon ablation evaluated by the procedure time, amount of contrast agent used, and number of freezes);(b)Patient dependency (demographics/comorbidities, type of atrial fibrillation, CHA_2_DS_2_VASc score, electrocardiography data, echocardiography data, laboratory values, and medication intake).

### 2.3. Microembolic Signal Detection Using Transcranial Doppler Ultrasonography

MESs in the arterial circulation system produce a brief visible and audible signal within the regular Doppler frequency spectrum derived across the middle cerebral artery, allowing their detection using transcranial Doppler ultrasonography [24]. From a physical point of view, this is made possible by the strong reflection of the ultrasound at the interface between the embolus and the surrounding blood. This creates an audible and visible impedance jump within the derived frequency spectrum. In several studies, it was possible to prove that the signals derived from transcranial Doppler ultrasonography are caused by “gaseous” or “solid” emboli measuring approximately 5 μm (gaseous) to 100 μm (solid) [25,26]. In the vast majority of cases, these are thought to remain clinically silent [15].

In our study, noninvasive transcranial Doppler ultrasonography was performed using DWL MultiDop T (DWL Elektronische Systeme GmbH, Sipplingen, Germany) to display the flow signal over the arteria cerebri media (Figure 3A). Methodological aspects of MES detection using transcranial Doppler Ultrasound were performed in accordance with internationally accepted recommendations [27]. Briefly, immediately before the start of the procedure, the transcranial transducer was fixed to the temporal bone after the best acoustic window had been located—with a bilateral derivation of the middle cerebral artery Doppler spectrum if possible. The sample gate was 8 mm, and the Doppler gain was reduced. Signal recording started immediately before the inguinal puncture and ended again when the inguinal sheaths were removed. With regard to MESs, a distinction was made between (1) a single MES or a HITS (=High-Intensity Transient Signal) with an acoustic and visual uni- or bidirectional signal and an impedance > 8 dB above the baseline (Figure 3B,C) and (2) showers, which were defined as HITSs that could not be counted individually, representing pronounced embolizations (Figure 3D). By definition, “solid” emboli were strictly unidirectional within the Doppler spectrum and had an acoustic impedance > 8 dB over the baseline (Figure 3B). “Gaseous” emboli were large, high-intensity bidirectional signals exceeding the Doppler spectrum (Figure 3C). Because the size and composition of an embolus influence the intensity of an MES independently of each other, it is sometimes difficult or impossible to classify a detected MES as a clearly “gaseous” or “solid” embolus. Therefore, in addition to the detected MES, another parameter was included in our study to increase the accuracy in assessing the entity detected and defining the embolus as gaseous or solid: the moment the MES occurs. This was tracked by means of a chronological written log of each step of the procedure. To this end, the clock of the DWL MultiDop T device was synchronized with the clock of the electrophysiological registration unit (BARD, Boston Scientific, Marlborough, MA, USA). In order not to influence the electrophysiologists and the assistance team involved directly in the ablation procedure, the acoustic signals for MES detection were muted. In addition, the screen showing the MES was not visible to the ablation team. In case of a loss of the detection signal during the continuous recording of the flow signal over the arteria cerebri media as a result of slippage of the ultrasound probe on the patient’s temple, repositioning was carried out immediately by a physician who was not involved in the ablation procedure.

At the end of the procedure, all HITSs and showers were counted manually and assigned to the various steps of the ablation procedure based on the time log kept during the procedure and categorized as either “gaseous” or “solid” emboli. The evaluation was carried out by two different neurologists experienced in the use of the TCD technique and cerebral MES detection. Both were blinded to each other as they worked, and each neurologist worked with an electrophysiologist familiar with the cryoballoon ablation technique. In case of dissent, a consensus read involving all four medical colleagues was undertaken. After discussion, a consensus judgment was documented and considered for further analysis.

### 2.4. Neuropsychological Testing

All baseline and follow-up neuropsychological assessments were performed by applying the German Plus version of the “Consortium to Establish a Registry for Alzheimer’s Disease Neuropsychological Assessment Battery” [28,29]. This test battery is not used solely for demonstrating deficits in dementia patients but also for assessing cognitive decline in other circumstances, such as heart interventions [30,31,32,33]. The standard version contains an evaluation of verbal fluency, a modified version of the Boston Naming Test, global cognition (Mini-Mental State Examination—MMSE), verbal memory, and construction practice and delayed recall [34]. The Plus version addresses three additional items related to the processing speed (Trail Making Test A—TMT A) and executive/frontal lobe functioning (Trail Making Test B—TMT B, letter fluency: S-words) [29]. The Trail Making Test B/A ratio (TMT B/A) was calculated as an additional measure of executive function [35]. In each study patient, the CERAD Plus battery was performed three times: prior to the ablation procedure, 1 day after, and 3 months later. Tasks were performed in the appropriate order:Verbal fluency (animals): Measures disturbances in verbal production and examines semantic memory and languageBoston naming test: Assesses the patient’s verbal ability to name line drawingsMMSE total score: Tests spatial orientation problems, memory, attention, arithmetic, and languageWord list learning sum: Assesses how well patients retain newly learned informationConstructional praxis: Examines constructive practiceWord list recall: Tests the ability to remember a list of words viewed previouslyWord list visual recall: Tests verbal memory, delayed verbal memory, recognition, and recall versus memory deficitsConstructional praxis recall: Tests nonverbal memory, delayed figural memory, and free reproductionTrail making test A: Measures psychomotor speedTrail making test B: Reflects the performance of executive functionsVerbal fluency (s-words): Tests verbal fluency in a strategy-oriented manner

All examinations were carried out by two physicians trained for this purpose in accordance with the known specifications [28]. The subtests of the CERAD Plus battery were grouped into four domains: executive, memory, language, and visuospatial. The CERAD overall score was calculated as previously published [36].

### 2.5. Atrial Fibrillation Ablation Procedure

All procedures were performed under conscious sedation with diazepam (Ratiopharm, Germany) and piritramide (Hameln, Germany). After double puncture of the right femoral vein using the Seldinger guidewire technique and insertion of a 7 F and 8.5 F sheath, a decapolar catheter (ViaCath^®^, Biotronik, Berlin, Germany) was selectively placed in the HIS and then in the CS position. A single fluoroscopic, pressure-guided trans-septal puncture was followed by a SL-1 sheath (Abbott, St. Paul, MN, USA) using a modified Brockenbrough technique (BRK-1, Abbott). Left atrium (LA) and pulmonary vein (PV) anatomies were visualized by means of PV angiography under rapid pacing via a decapolar catheter in the right ventricular apex with a heart rate of 300 ms for approximately 5 to 10 s. During angiography via the SL-1 sheath, 15 mL of contrast agent (Xenetics^®^, Guerbet, Germany) was administered manually for both the right and left upper pulmonary veins. Afterwards, an exchange wire was placed preferably in the left, but alternatively in the right, superior PV, and the SL-1 sheath was replaced with a steerable 15 F Cryo-Advance sheath (Medtronic Inc., Mounds View, MN, USA). The LA and PV borders, including the PV ostia and trans-septal puncture site (determined using PV angiography), were marked on the monitor screen to ensure safer movement and positioning of the cryoballoon during the ablation procedure. PV signals were mapped with an inner lumen spiral mapping catheter (Achieve catheter^TM^, 20 mm diameter, Medtronic) before, during, and after each cryoenergy application. Guided by the Achieve^TM^ catheter, the 28 mm cryoballoon was exclusively advanced through the sheath into the LA, inflated proximally to the PV ostium, and then gently pushed to seal off the PV. Vessel occlusion and atrial backflow were evaluated through the selective injection of contrast medium via an automatic injection pump (CVi^TM^, ACIST Europe B.V.) standardized with a volume of 5 mL, flow of 3 mL/s, and pressure of 350 psi. After the best possible occlusion was achieved, a 240 s freeze–thaw cycle was performed. If minus 60 °C was reached before the 240 s freeze-thaw cycle had expired, the freeze time was stopped earlier. During the ablation of the septal PVs, the decapolar catheter was positioned in the superior vena cava for ipsilateral phrenic nerve pacing. Phrenic nerve function was monitored using diaphragmatic compound motor action potentials (CMAPs) [37]. The ablation was terminated immediately if the CMAPs decreased substantially, indicating weakening or loss of diaphragmatic contraction. An additional left atrial roof ablation was performed in all patients with persistent atrial fibrillation, as described above [13]. Briefly, sequential overlapping 180 s freezes were applied (starting near the position used for left superior PV isolation) along the left atrial roof by means of a slight clockwise rotation combined with slight sheath retraction and incremental advancement of the cryoballoon until the position used for right superior PV isolation was reached. The Achieve catheter was anchored in the left superior PV for all cryoballoon positions at the left atrial roof. In AF patients without intraprocedural conversion to SR, electrical cardioversion was performed, and PVI and a conduction block at the left atrial roof were then verified, as previously described [12]. Patients with documented typical right atrial flutter or periprocedural typical atrial flutter prior to the procedure received an additional cavotricuspid isthmus radiofrequency (RF) ablation (Alcath Black Flux^®^, Biotronik, Berlin, Germany). Pericardial effusion was excluded by echocardiography immediately after ablation. After removal of the venous sheaths, the access site was closed with subcutaneous temporary purse-string sutures [38]. Afterwards, patients were monitored telemetrically for ≥24 h.

To examine the factors influencing the investigatory dependency in relation to the frequency and characterization of MESs, the ablations in the study were carried out by 3 electrophysiologists with different levels of experience in cryoballoon ablation:

Investigator 1: >50 to ≤100 cryoballoon ablations

Investigator 2: >100 to ≤500 cryoballoon ablations

Investigator 3: >500 cryoballoon ablations

The number of ablation procedures was divided equally between the three electrophysiologists where possible.

### 2.6. Anticoagulation Management

The consumption of direct oral anticoagulants (DOAC) was only paused on the morning of the ablation procedure. DOAC intake was resumed in adequate doses two hours after the procedure, provided that pericardial effusion had been ruled out and the inguinal puncture site was unremarkable. In case of phenprocoumon intake, the procedure was performed under an INR of 2.0–3.0. Transesophageal echocardiography was performed in the case of insufficient oral anticoagulant intake within the 4 weeks before the procedure and in all cases of persistent atrial fibrillation. Periprocedurally, heparin was administered intravenously with a target activated clotting time (ACT) of ≥300 s. No protamine was given at the end of the procedure.

### 2.7. Gastroesophageal Endoscopy

Due to the anatomical proximity of the left atrium to the esophagus, there is a risk of thermal injury to the esophagus during atrial fibrillation ablation procedures. The most serious complication here is the development of an esophagoatrial fistula, which is fatal in most cases. The progression from a thermal esophageal lesion to an esophagoatrial fistula has been described with a frequency of up to 0.2%, although the number of unreported cases is probably higher [39]. Endoscopically detected esophageal thermal lesions (EDEL) have been reported in up to 22% of patients following cryoballoon ablation [40]. Therefore, additional cryoballoon ablation techniques beyond pulmonary vein isolation, such as LARA, raise concerns of a possible increased risk of EDEL. In the light of this knowledge, we established an internal procedure to test for EDEL in patients undergoing ablation in order to unveil silent pathologies and prevent subsequent complications. When our study participants underwent endoscopic examination, data regarding esophageal lesions were systematically recorded offline and used for further analyses. This meant that we were able to investigate factors potentially associated with the burden of MES, e.g., the incidence and characterization of thermal injuries after PVI only (Group A) and beyond (Group B). Gastroesophageal endoscopic examination was performed in all patients 24 to 36 h after the ablation procedure following the second CERAD Plus examination. According to the current recommendations, esophageal lesions were classified based on the novel Kansas City classification [39].
Type 1: erythemaType 2a: superficial ulcerType 2b: deep ulcerType 3a: perforation without communication with the atriaType 3b: perforation with atrioesophageal fistula


### 2.8. Statistical Analysis

#### 2.8.1. Statistical Study Planning and Sample Volume Calculation

Given that we were able to recruit 100 patients for inclusion in this study, the goal of identifying patients with pronounced embolization, in particular, appears realistic. The case number calculation for this study was based on data from Von Bary et al. [23]. The authors detected showers in 12 of 35 patients (34%) during the procedure. Taking this relative frequency as a basis, we identified 34 patients (34%) with a power of 80% (1-ß, ß = errors of two kinds) and an error of the first kind of α = 5%. In this case, the confidence interval for the relative frequency examined would extend from 24% to 43% over a maximum of 19%.

#### 2.8.2. Current Analysis

For the current presentation, all data for continuous variables were presented as median and interquartile ranges. Categorical variables were reported as frequencies and percentages. The normal distribution was verified using Kolmogorov–Smirnov’s one-sample test. Nonparametric data were analyzed by applying a two-tailed Mann–Whitney U-test. Fisher’s exact test was used to compare relative frequencies. Statistical analyses were performed using SPSS software (version 22.0, IBM Corporation, Armonk, NY, USA).

## 3. Results

From February 2021 to August 2022, of the 167 screened outpatient clinic patients assigned for atrial fibrillation ablation using cryoballoon, 100 were included in this prospective observational study: Group A with paroxysmal atrial fibrillation (*n* = 50) and Group B with persistent atrial fibrillation (*n* = 50) (Figure 2). The main reason that patients were excluded from the study was the lack of a transcranial bone acoustic window (*n* = 55; 33%). Other reasons for study exclusion were the refusal to participate (*n* = 9; 5%) or patient expectation that they would miss the three-month follow-up visit (*n* = 3; 2%). In the end, 91 out of 100 patients fully completed the study. Nine patients failed to attend the 3-monthsfollow-up visit, 6 in Group A and 3 in Group B. After telephone consultation with each of these nine patients, it was determined that the routine follow-up after PVI had already been performed by the resident cardiologist, and these patients felt that taking part in the last CERAD Plus testing would be inconvenient. Ultimately, all 9 patients were in sinus rhythm and free from any symptoms (evidenced by a 24-h long-term ECG on an outpatient basis and the information provided by each patient).

The median age of the total study population was 65.5 years with a median CHA_2_DS_2_-VASc score of 2. Patient characteristics are summarized in Table 5. Viewed by group, collectives differ in terms of sex distribution, body mass index, atrial size, left ejection fraction, and NT-Pro BNP levels. Procedural characteristics are displayed in Table 6. Acute pulmonary vein isolation was achieved in all patients, with a median number of 5 freeze applications. A complete conduction block of the LARA was achieved in 82% of patients in Group B. All four documented phrenic nerve injuries were transient and recovered fully within 24 h to 3 months. None were associated with LARA.

## 4. Discussion

To the best of our knowledge, this study includes the largest exclusive cryoballoon technique patient group ever examined, in which the frequency, characterization, and factors influencing microembolizations in the cerebral circulation were examined, especially with regard to their neuropsychological effects. Furthermore, this is the first study to investigate this issue in patients who have received cryoballoon ablation beyond pulmonary vein isolation.

While it is known that cerebral microembolizations occur frequently after atrial fibrillation ablation and correspond to small microinfarcts in magnetic resonance imaging (MRI), their clinical significance is unclear [16]. Short-term follow-up using MRI has already shown that these cerebral lesions are no longer detectable in over 90% of all cases [41].

On the other hand, we do not know for sure whether microembolizations that do not cause cerebral image morphological changes in MRI potentially have a clinical impact and/or an influence on neuropsychological abilities in ablated AF patients. As demonstrated in previous clinical investigations with comparable circumstances, microembolic infarctions do not possess clinical relevance and are often interpreted as a surrogate marker [15]. We therefore focused our investigation on clinical parameters such as neuropsychological ability and clinical status.

Cryoballoon ablation beyond PVI is routinely rarely used worldwide, which explains why no data are available to date on microembolization and its neuropsychological outcomes. The alleged lack of precise control of the cryoballoon in the left atrium “outside” the pulmonary vein ostia unsettles many users, so this technique is not used routinely despite some promising data [42]. In our clinical routine, as well in our present study, we create a roofline without any complications in patients with persistent atrial fibrillation without the use of a 3D system. This is achieved by displaying the anatomical conditions by simply drawing the contours of the left atrium, the pulmonary veins, and the trans-septal puncture site directly on the examiner’s screen using a board marker, based on previous publications [12,13]. The frequency of a successful LARA conduction block in our study is comparable to that presented in previously reported data [12,13].

The patients examined in this study represent a typical patient collective with paroxysmal and persistent atrial fibrillation in terms of age, sex distribution, and comorbidities and are comparable with collectives from other atrial fibrillation studies [12,13,23]. For this reason, only one-third of the screened patients had to be excluded from the current study due to a missing transcranial bone acoustic window. It has been described that a transcranial bone acoustic window can be missing in up to 30% of patients examined [43,44,45]. Female sex and increasing age are associated with an insufficient acoustic bone window [43,44,45].

## 5. Conclusions

This study managed to collect data from the largest case series to date focusing on cerebral microembolizations during pulmonary vein isolation and beyond using the cryoballoon technique exclusively. The data collected make it possible to address questions regarding the frequency, characterization, and factors influencing microembolizations and their clinical impacts on the neuropsychological field.

## Figures and Tables

**Figure 1 diagnostics-13-01660-f001:**
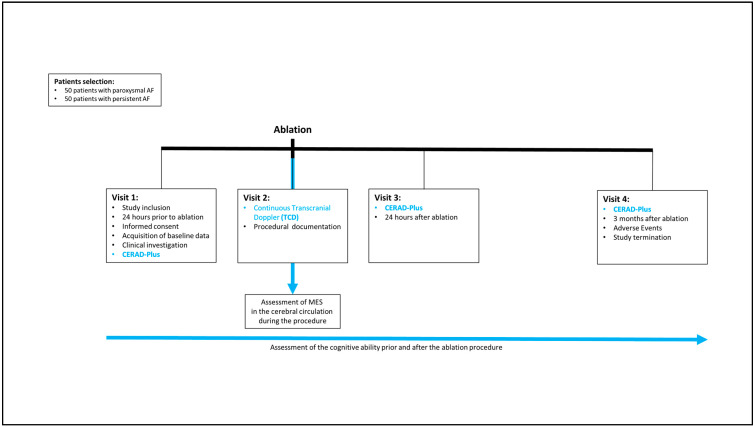
Study flowchart.

**Figure 2 diagnostics-13-01660-f002:**
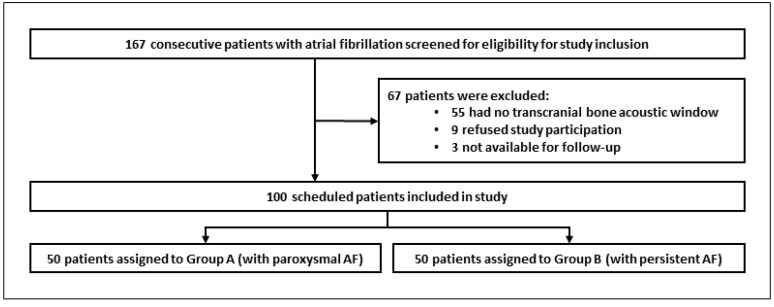
Flowchart of patients included. ‘AF’= Atrial fibrillation. Group A: paroxysmal atrial fibrillation—receiving cryoballoon PVI only. Group B: persistent atrial fibrillation—receiving cryoballoon PVI and additional left atrial roof ablation (LARA).

**Figure 3 diagnostics-13-01660-f003:**
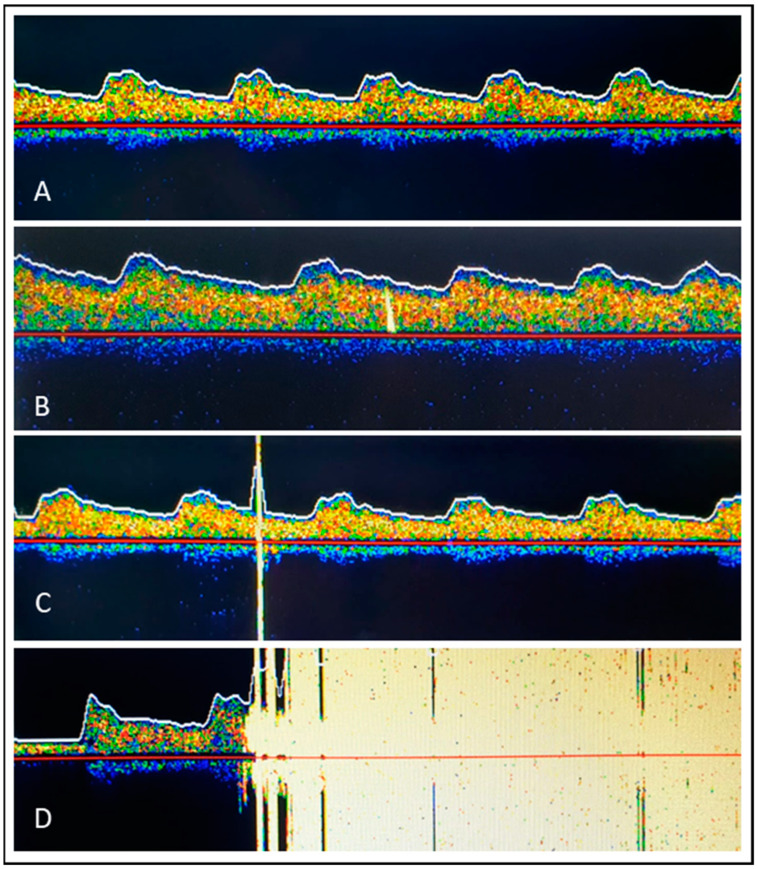
(**A**): Normal flow profile over the middle cerebral artery; (**B**): Single microembolic signal as presented by one high-intensity transient signal (HITS); Single strictly unidirectional signal within the Doppler spectrum implying a “solid” embolus; (**C**): Large, high-intensity bidirectional signal (HITS) exceeding the Doppler spectrum implying a “gaseous” embolus; (**D**): Many (innumerable) HITSs, indicating a shower.

**Table 1 diagnostics-13-01660-t001:** Inclusion criteria.

Age	≥18 years
Atrial fibrillation stage	paroxysmal or persistent
Indication for ablation therapy	under current atrial fibrillation guidelines
Transcranial bone acoustic window	available unilateral or bilateral
Written informed consent	to be provided prior to enrollment

**Table 2 diagnostics-13-01660-t002:** Clinical data collected at time of study inclusion.

Demographics (age, sex, body mass index)
Type of atrial fibrillation (paroxysmal vs. persistent)
CHA_2_DS_2_VASc score
Electrocardiography data
Echocardiography data
Laboratory blood values
Comorbidities
Current medication
Neuropsychological baseline status (1. CERAD Plus test)
Available detection site of cerebral microembolic signals

**Table 3 diagnostics-13-01660-t003:** Periprocedurally collected study data.

Frequency and Characterization of Microembolic Signals over Time:
- Venous access until TSP
- TSP
- PV/LA Angiography
- Cryoballoon introduction into LA
- PV occlusion, freeze cycle, and thawing phase for each PV
- Left atrial roof ablation (LARA) freeze cycles
- Sheath retreat to the RA until inguinal suture
EP procedure data:
- Procedure time
- Fluoroscopic time
- Radiation dose
- Acute ablation success
- Cryoenergy application time
- Number of freezes (PV/roofline)
- Nadir temperature
- Number of isolated PVs
- Number of blocked LARA
- Amount of contrast agent
- Periprocedural complications
- Investigator dependency

‘TSP’ = transseptal puncture; ‘PV’ = Pulmonary Vein; ‘LA’ = left atrium; ‘RA’ = right atrium; ‘EP’ = electrophysiology. ‘LARA’ = left atrial roof ablation.

**Table 4 diagnostics-13-01660-t004:** Postprocedural follow-up data within 3 months.

Neuropsychological testing 24 h after PVI
Endoscopic esophageal examination 24 to 36 h after PVI, after 2. CERAD Plus test
Neuropsychological testing 3 months after PVI
Evaluation of mid-term complications
Cardiac rhythm evaluation by electrocardiography (resting and 24-hour Holter ECG)
Current medication

‘PVI’ = pulmonary vein isolation; ‘ECG’ = electrocardiography.

**Table 5 diagnostics-13-01660-t005:** Baseline characteristics.

	Total (*n* = 100)	Paroxysmal AF (*n* = 50)	Persistent AF (*n* = 50)	*p*
Age (median, IQR) (years)	65.5 (58–72)	66 (57–71)	65.5 (58.8–74)	0.564
Sex				
Male	69 (69%)	28 (56%)	41 (82%)	0.009
Female	31 (31%)	22 (44%)	9 (18%)	
BMI (median, IQR), (Kg/m^2^)	28.45 (25.5–33.35)	27.4 (23.9–31.8)	30.3 (26.7–34.7)	0.016
CHA_2_DS_2_VASc (median, IQR, range)	2 (1–3, 0–7)	2 (1–3)	2 (1–4)	0.056
Echocardiography				
Left atrium diameter (cm) (median, IQR)	4.17 (3.6–4.6)	3.8 (3.2–4.2)	4.6 (4.1–5.0)	<0.001
Left atrium volume (ml) (median, IQR)	81 (60–102)	62 (48–83)	94 (76.5–120.5)	<0.001
Left atrium index (ml/m^2^) (median, IQR)	39 (29.9–50.9)	31.7 (24–38.9)	48.1 (38.5–56.1)	<0.001
Mitral insufficiency I–II° (median, IQR, range) (*n* = 77)	I° (I°–I°; I°–II°)	1 (0–1)	1 (1–1)	0.037
Left ejection fraction (median, IQR, range) (%)	60 (55–60; 40–75)	60 (60–65)	55 (50–60)	<0.001
Comorbidities				
Hypertension	71 (71%)	33 (66%)	38 (76%)	0.387
Diabetes mellitus	18 (18%)	7 (14%)	11 (22%)	0.378
LDL (mg/dL) (median, IQR)	117 (86.5–153.5)	111 (77–155)	114 (97–148)	0.613
Coronary artery disease	21 (21%)	10 (20%)	11 (22%)	0.999
Sleep apnea	11 (11%)	3 (6%)	8 (16%)	0.201
Heart insufficiency				
NT-Pro-BNP (ng/L) (median, IQR) (*n* = 61)	417 (86.5–153.5)	232.5 (138.3–470.5)	742.0 (206.5–2178.5)	0.004
Kidney disease				
eGFR (ml/min/1.73 m^2^) (median, IQR)	76 (64.4–88.9)	77.5 (66.8–89.1)	75.2 (58.4–89.0)	0.295
Gastroesophageal reflux disease	5 (5%)	3 (6%)	2 (4%)	0.999
Previous stroke	8 (8%)	2 (4%)	6 (12%)	0.269
Severe carotid artery stenosis	1 (1%)	0	1 (2%)	>0.999
Carotid plaque without hemodynamic relevance	6 (6%)	2 (4%)	4 (8%)	0.678
Medication prior procedure				
Intake of oral anticoagulants	95 (95%)	45 (90%)	50 (100%)	0.056
Intake of platelet inhibitors	7 (7%)	1 (2%)	6 (12%)	0.112
Intake of proton pump inhibitors	25 (25%)	12 (24%)	13 (26%)	>0.999
Intake of sedative drugs	14 (14%)	7 (14%)	7 (14%)	>0.999

‘AF’ = atrial fibrillation; ‘IQR’ = interquartile range; ‘LDL’ = low-density lipoprotein; ‘eGFR’ = estimated glomerular filtration rate; ‘BMI’ = body mass index.

**Table 6 diagnostics-13-01660-t006:** Procedural characteristics.

	Total (*n* = 100)	Paroxysmal AF (*n* = 50)	Persistent AF (*n* = 50)	*p*
Procedure time (min) (median, IQR)	110 (92–128)	96 (84.5–114.0)	118.5 (108.8–130.3)	<0.001
Fluoroscopic time (min) (median, IQR)	26.5 (15.4–26.5)	18.2 (11.5–23.9)	23.2 (17.9–30.3)	0.002
Radiation dose (uGym^2^) (median, IQR)	1001 (691.3–1485.3)	776 (569.7–1516.5)	1318.5 (970–1613.8)	<0.001
Amount of contrast agent (mL) (median, IQR)	63 (52–80)	60 (50–80)	67 (55–80)	0.113
Additional right isthmus ablation	11 (11%)	7 (14%)	4 (8%)	0.525
Pulmonary vein isolation				
Acute ablation success	100 (100%)	50 (100%)	50 (100%)	0.999
Total number of freezes (median, IQR)	5 (4–5)	5 (4–5)	5 (4–6)	0.172
Cryoenergy application time (min) (median, IQR)	16.16 (15.12–20.0)	16 (15.2–19.5)	17.1 (15–22.1)	0.435
Nadir temperature				
Nadir temperature RSPV (°C) (median, IQR)	−53 (−56 to −49)	−54 (−56 to −51)	−52 (−56 to −48)	0.326
Nadir temperature RIPV (°C) (median, IQR)	−51 (−55 to −47)	−52 (−55 to −47)	−50 (−54 to −47)	0.323
Nadir temperature LSPV (°C) (median, IQR)	−48 (−53 to −45)	−48 (−54 to −44)	−49 (−53 to −46)	0.218
Nadir temperature LIPV (°C) (median, IQR)	−46 (−51 to −45)	−46 (−48 to −45)	−48 (−54 to −45)	0.062
Left Atrial Roof Ablation (LARA)				
Acute ablation success	91 (91%)	50 (100%)	41 (82%)	0.003
Total number of freezes (median, IQR, range)	0 (0–4)	0 (0–0)	4 (3–4; 3–6)	<0.001
Cryoenergy application time (min) (median, IQR)	0 (0–720)	0 (0–0)	9 (9–18)	<0.001
Nadir temperature (°C) (median, IQR)	0 (−39 to 0)	0 (0–0)	−40 (−33 to −46)	<0.001
Acute complications (total)				
Phrenic nerve injury	4 (4%)	3 (6%)	1 (2%)	0.617
Pericardial effusion	0 (0%)			
Stroke/TIA	0 (0%)			
Vascular groin complication	0 (0%)			

‘AF’ = atrial fibrillation; ‘PV’ = pulmonary vein; ‘RSPV’ = right superior pulmonary vein; ‘RIPV’ = right inferior pulmonary vein; ‘LSPV’ = left superior pulmonary vein; ‘LIPV’ = left inferior pulmonary vein; ‘LA’ = left atrium; ‘TIA’ = transient ischemic attack.

## Data Availability

The data presented in this study are available on request from the corresponding author. The data are not publicly available due to privacy and ethical restrictions.

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
