# Peer review of "Microembolizations in the Arterial Cerebral Circulation in Patients with Atrial Fibrillation Ablation Using the Cryoballoon Technique—Protocol and Methodology of a Prospective Observational Study"

_diagnostics, 2023, doi:10.3390/diagnostics13091660_

Round 1

Reviewer 1 Report

In the study by Erkapic D, “Microembolizations in the arterial cerebral circulation in patients with atrial fibrillation ablation using the cryoballoon technique – protocol and methodology of a prospective observational study”

In general, patients with atrial fibrillation have poorer cognitive function than patients without atrial fibrillation. This can be due to loss of atrial contraction and reduced cardiac output. If AF ablation induces functional remodeling of atrium, it may improve cognitive function in AF patients. However, there have been many reports of mircroembolism after AF ablation, thus if the ablation itself reduces cognitive function, it would defeat the purpose.

This study is a unique study that focuses on the problem and is very interesting. This is the protocol of their study.

There are some serious defects must be addressed.

Concern #1

You said "additional cryoballoon left atrial roof line and posterior wall isolation have proven beneficial in achieving an arrhythmia-free outcome in patients with persistent atrial fibrillation" (page 1, line 35-38) What do you mean ”proven" ?

In the RCT of posterior wall isolation named the CAPLA Randomized Clinical Trial, outcome of AF free survival in additional posterior wall isolation on PVI was still controversial. Please tell me about positive data of posterior wall isolation in large-scale RCTs or systematic meta-analysis.

Concern #2

In your study, participants were divided into a persistent atrial fibrillation group and a paroxysmal atrial fibrillation group. Furthermore, the methods of catheter ablation differed between the two groups.

Your hypothesis was unclear in comparing these two groups.

Please define your hypotheses and object in order to eliminate the multiplicity issue.

Concern #3

In many previous study, MRI have been used to assess microembolism after AF ablation as you described in the article. By the way, in your study, 33% of patients were difficult to assess using transcranial Doppler ultrasound. Please give a rationale for using this less successful method instead of MRI.

Concern #4

There seems to be less information on the patients' characteristics.

Has the proportion of patients with carotid artery stenosis or carotid plaque, which is expected to have a strong impact on cerebral blood flow, been investigated? What is percentage of patients using tranquilizer and sleeping pills. Those medications can affect therr cognitive function.

Author Response

Point by point response to reviewers comments

Reviewer #1

We thank the reviewer for his/her constructive comments and suggestions, which clearly strengthened our manuscript. We have addressed all suggestions as follows:

  1. You said "additional cryoballoon left atrial roof line and posterior wall isolation have proven beneficial in achieving an arrhythmia-free outcome in patients with persistent atrial fibrillation" (page 1, line 35-38) What do you mean ”proven" ?

In the RCT of posterior wall isolation named the CAPLA Randomized Clinical Trial, outcome of AF free survival in additional posterior wall isolation on PVI was still controversial. Please tell me about positive data of posterior wall isolation in large-scale RCTs or systematic meta-analysis.

We agree with the reviewer that there are no large-scale randomized clinical trials or systematic meta-analysis which have shown advantages regarding arrhythmia free survival after extrapulmonary ablation strategies as an add-on to pulmonary vein isolation. To avoid mis-understandings we deleted the word “proven” and changed this passage since we did not even investigate posterior wall isolation (please see page 3, lines 7-9). Furthermore, we removed the associated references from the manuscript. We thank you for pointing this out.

  1. In your study, participants were divided into a persistent atrial fibrillation group and a paroxysmal atrial fibrillation group. Furthermore, the methods of catheter ablation differed between the two groups.

Your hypothesis was unclear in comparing these two groups.

Please define your hypotheses and object in order to eliminate the multiplicity issue.

The attempt to include 50:50 AF patients with persistent AF versus paroxysmal AF intended to homogenize the study population. This was one of the points in previous studies, colleagues investigated randomly heterogenous populations with consecutive patients.  The main purpose of the presented study was to investigate the burden of cerebral Microembolic Signals (MES) during the procedure; this was conducted in a cross-sectional design. Using this as starting point we investigated the influence of the procedure on the cognitive ability of the individual patient; including the entire cohort (longitudinal cohort study design), the main objective was to determine if generally the procedure results in a cognitive decline proximate to the procedure and in mid-term follow up; The hypothesis was here, if the procedure affects the cognitive performance. As a subgroup analysis there is beneficial to analyze separately effects in the subgroup of persistent versus paroxysmal AF patients; in this case the number of 50 patients each per subgroup would surpass previous studies addressing this topic. In a further step we intend to prove a potential correlation between a cognitive decline and the burden of microembolization.

In conclusion the following hypotheses/ objectives could be summarize:

  1. Burden of cerebral microembolization during the procedure, which is conducted as described in the section methods (cross-sectional design).
  2. Assessment of a potential cognitive decline caused by the procedure (longitudinal cohort study design).
  3. Analysis of a potential causative correlation between burden of cerebral microembolization and cognitive decline.
  4. Investigation of factors potentially associated with cognitive decline after the procedure, burden of cerebral microembolization, the occurrence of adverse events during the procedure (esophageal lesions). Among these factors such as the type of AF, baseline data (e.g. sex), or vascular risk factors could be mentioned here.           

 Substantial changes were made to our manuscript to highlight our study design and our objectives in accordance with the reviewers recommendations. Please see pages 5-7 (Section: Study design and objectives). Furthermore we added a new Figure 1 (study flow chart) to the manuscript.

  1. In many previous study, MRI have been used to assess microembolism after AF ablation as you described in the article. By the way, in your study, 33% of patients were difficult to assess using transcranial Doppler ultrasound. Please give a rationale for using this less successful method instead of MRI.

Magnetic resonance cerebral imaging shows us image morphological changes in the brain tissue after atrial fibrillation ablation. Unfortunately, it does not allow us to detect the frequency and type of microembolizations that occur during atrial fibrillation ablation. Microembolic signal (MES) detection using transcranial Doppler ultrasonography has the potential to identify the embolic process from several potential sources of embolism during pulmonary vein isolation. For this reason non-invasive transcranial Doppler ultrasonography was used, which is a long-standing standardized and established diagnostic tool in Neurology. In addition, we know from previous studies that the image-morphological brain changes detected by MRI are mostly of a temporary nature and have not yet led to clearly neuropsychological impairments.  On the other hand, we do not know for sure whether microembolizations that do not cause cerebral image morphological changes do have possible impact on the neuropsychological abilities in ablated AF patients.  Therefore, in our study the focus was placed on the MES detection direct during the procedure and the crucial excessive neurophysiological testing. 

We thank the reviewer for his/her comment. We point this out in the discussion section (page 17, lines 7-16). Furthermore we have made the capabilities of the transcranial Doppler ultrasonography more understandable in the Section: “Microembolic Signal Detection using Transcranial Doppler Ultrasonography” (page 7 lines 18-25, page 8 lines 1-26 and page 9 lines 1-14). In addition we renewed Figure 3.  

  1. There seems to be less information on the patients' characteristics. Has the proportion of patients with carotid artery stenosis or carotid plaque, which is expected to have a strong impact on cerebral blood flow, been investigated? What is percentage of patients using tranquilizer and sleeping pills. Those medications can affect their cognitive function.

We thank the reviewer for his/her helpful hint. Considering the two groups examined, we have replaced Table 5 in the manuscript with a more detailed table regarding patients` characteristics. In this you will also find the number of patients with carotid plaques and stenoses as well as patients who have taken the drugs mentioned above. Furthermore we added to the manuscript the procedural data (see Table 6).  

Reviewer 2 Report

The authors present the study in which they focus on cerebral microembolization during the ablation for atrial fibrillation.

To my opinion there is a lack of data describing the cerebral microembolization (definition, mechanisms of development, severity, impact on outcome and importance). The aim of the study is not enough justified, it is worth to emphasize the importance of the problem. 

It is unclear what the authors meant writing about secondary endpoints. What are technical variables and individual factors? 

The authors present the baseline clinical characteristics of patients. It is reasonable to compare the groups (persistent vs paroxysmal AF).

Also, it is said that the 3-months visit has been completed. May be, it is worth to present the full results of the study. 

From the clinical point of view it is clear why the gastroesophageal endoscopic examination was performed, but I would recommend to explain it in the text in more details (how often the esophageal lesion occurs, what impact it has, et.c.).

I would recommend to think about the structure of the manuscript (the paragraph describing the micro embolism somewhere after the endoscopic examination). 

Author Response

Point by point response to reviewers comments

Reviewer #2

We thank the reviewer for his/her constructive comments and suggestions, which clearly strengthened our manuscript. We have addressed all suggestions as follows:

  1. To my opinion there is a lack of data describing the cerebral microembolization (definition, mechanisms of development, severity, impact on outcome and importance). The aim of the study is not enough justified, it is worth to emphasize the importance of the problem.

 Substantial changes were made to our manuscript to highlight our study design and our objectives in accordance with the reviewers recommendations. Please see pages 5-7 (Section: Study design and objectives). Furthermore we added a new Figure 1 (study flow chart) to the manuscript and renewed the Section “Microembolic Signal Detection using Transcranial Doppler Ultrasonography” (please see pages 7-9). In addition we updated Figure 3.   

  1. It is unclear what the authors meant writing about secondary endpoints. What are technical variables and individual factors?

The secondary endpoints in this study include the identification of factors favoring the occurrence of microembolizations. On the one hand, these are technical variables that relate to the electrophysiological procedure, as listed step by step in Table 3: venous access, transseptal puncture, PV/LA angiography, Cryoballoon introduction into the LA, PV balloon occlusion, freeze cycle and thawing phase for each PV, LARA freeze cycles and sheath retreat to the RA.   On the other hand, these are also individual factors as the investigator dependency (influence of the examiner´s experience in cryobaloon ablation evaluated by procedure time, amount of contrast agent, number of freezes) as well as patients individual factors as listed in Table 2 and 3: demographics/comorbidities, type of atrial fibrillation, CHA2DS2VASc score, echocardiography data, laboratory values and medication intake.For a better understanding of the technical and individual factors, we have mentioned them separately in the study design and objectives section (please see page 7, lines 4-15).  

  1. The authors present the baseline clinical characteristics of patients. It is reasonable to compare the groups (persistent vs paroxysmal AF).

We thank the reviewer for his/her hint. Considering the two groups examined, we have replaced Table 5 in the manuscript with a more detailed table regarding patients` characteristics.

  1. Also, it is said that the 3-months visit has been completed. May be, it is worth to present the full results of the study.

Due to the size of the data volume, the full results of the study will be presented in a separate manuscript. To this manuscript we added the procedural data in total and per groups (see Table 6).

  1. From the clinical point of view it is clear why the gastroesophageal endoscopic examination was performed, but I would recommend to explain it in the text in more details (how often the esophageal lesion occurs, what impact it has, etc.).

Following the reviewer's recommendation, a corresponding passage was inserted into the manuscript (please see page 14 lines 1 to 25):

  1. I would recommend to think about the structure of the manuscript (the paragraph describing the micro embolism somewhere after the endoscopic examination).

We have revised the manuscript accordingly.                                                       

Round 2

Reviewer 1 Report

Author Damir Erkapic is to be congratulated for a nicely presented revised article.

The authors addressed my concerns sufficiently in this version.

I have no further comments.

Reviewer 2 Report

I would like to thank the authors for their responses and work with the manuscript. I do not have additional comments.